# Validation of the Palliative Care and Rapid Emergency Screening (P-CaRES) Tool in Germany

**DOI:** 10.3390/jcm14072191

**Published:** 2025-03-23

**Authors:** Julia Schmitz, Mitra Tewes, Baicy Mathew, Marie Bubel, Clemens Kill, Joachim Risse, Eva-Maria Huessler, Bernd Kowall, Maria Rosa Salvador Comino

**Affiliations:** 1Department of Palliative Medicine, University Hospital Essen, University of Duisburg-Essen, Hufelandstr. 55, 45147 Essen, Germany; julia.schmitz@ext.uk-essen.de (J.S.); mitra.tewes@uk-essen.de (M.T.); baicy.mathew@uk-essen.de (B.M.); marie.bubel@ext.uk-essen.de (M.B.); 2Center of Emergency Medicine, University Hospital Essen, Hufelandstr. 55, 45147 Essen, Germany; clemens.kill@uk-essen.de (C.K.); joachim.risse@uk-essen.de (J.R.); 3Institute for Medical Informatics, Biometry and Epidemiology at the University Hospital Essen, Hufelandstr. 55, 45122 Essen, Germany; bernd.kowall@uk-essen.de (B.K.)

**Keywords:** validation study, emergency service, palliative care, screening, assessment of healthcare needs

## Abstract

**Background/Objectives**: The “Palliative Care and Rapid Emergency Screening Tool” (P-CaRES) is used to identify patients with palliative needs in the emergency department. This study aims to translate, adapt, and validate the P-CaRES tool for the German healthcare context. **Methods**: This is a monocentric, non-interventional, retrospective study conducted in the emergency department of the University Hospital Essen, Germany. After a structured translation process, the tool’s face and content validity were evaluated using questionnaires completed by healthcare workers. Construct validity was established by comparing the results with the German-validated Supportive and Palliative Care Indicators Tool (SPICT). A palliative care physician blinded to the tool, independently assessed the screened patients, and made recommendations on the appropriateness of palliative care referrals. **Results**: Two hundred eighty-nine emergency department visits were examined. In addition, a total of 26 healthcare professionals were surveyed. 258 screenings were conducted, with P-CaRES identifying 50 positive cases (19.4%). Agreement between SPICT and P-CaRES was 88.4% (kappa: 0.67, 95% confidence interval (CI): [0.56, 0.78]), showing 64.7% sensitivity and 96.8% specificity. Expert comparison yielded 85.5% agreement (kappa: 0.54, 95% CI: [0.41, 0.68]), with 64.0% sensitivity and 90.8% specificity. Face and content validity showed strong agreement regarding the tool’s design, including its comprehensibility, applicability, ease of use, and precision, as well as its usefulness in referring patients to a specialized palliative care team. **Conclusions**: The study successfully validated a cultural and linguistic equivalent German version of the P-CaRES tool. Further research is necessary to assess the tool’s effectiveness in clinical practice.

## 1. Introduction

The benefits of an early integration of palliative care by palliative care services are far-reaching, including improvements in quality of life [1,2], mood [2], and symptom management, and [3,4] alleviating spiritual distress [5] while maintaining [1] or even prolonging [2] life expectancy. Furthermore, there is a reduction in hospital admissions [6,7], length of hospital and intensive care stays [8,9,10,11], and costs [3,12,13]. Earlier access to palliative care is associated with increased mortality rates within the home environment and greater utilization of hospice services [14,15].

Nevertheless, a significant number of patients with palliative care needs are either recognized late [16] or not at all [17], with delays often occurring between hospital admission and palliative care consultation [18]. According to the World Health Organization (WHO), only 10% of patients needing palliative care receive it [17].

Additionally, emergency department (ED) visits increase in the final months of life, with 81% of cancer patients visiting the ED in the last six months [19] and 34% in the last two weeks [7]. Of these, 72% are admitted to the hospital, with 77% of patients dying there and only 3% at home [7]. Similarly, and regarding non-cancer populations, Smith et al. [20] analyzed patients over 65 years old visiting the ED and found that approximately 75% visited the ED in the last 6 months and about 50% in the last month of life. Seventy-seven percent of these patients were admitted as inpatients, with 39% of them being admitted to an intensive care unit and 68% of all admitted patients dying while in the hospital.

There is a growing interest in objectifying the need for palliative care through prognostic indicators, patient circumstances, and clinical triggers. For instance, using an automated screening tool in the ED could help identify palliative care needs, provide a more comprehensive understanding of the prevalence of palliative patients in the ED, and highlight the disparity between resources and patient demand [21].

Several tools have been developed to assess palliative care needs in the ED, such as “SPEED” [22] and “PRIM-ER” [23], as highlighted in recent reviews [24,25]. However, there is no screening tool or algorithm for identifying possible palliative care needs in palliative patients in German-speaking countries, suitable for the time-critical ED environment. Existing tools, such as the Supportive and Palliative Care Indicators Tool (SPICT, Appendix A) [26,27,28] or the “IPOS” [29,30] and “MIDOS” [31] questionnaires, are not entirely applicable in the ED, as SPICT is lengthy and lacks cut-off values, while IPOS and MIDOS rely on patient self-assessment, although external assessment might be more appropriate in the ED context.

To ensure accurate assessments, tools must be rigorously validated [32,33]. The P-CaRES tool (Palliative Care and Rapid Emergency Screening Tool), developed and validated by George et al. [34,35] in the USA in 2015, is a validated, simple, and straightforward screening tool designed for the ED to identify patients needing early palliative care [36] (Figure 1).

This study aims to translate, adapt, and validate the P-CaRES tool to the Germanhealthcare context (Figure 2).

## 2. Methods

### 2.1. Patients and Study Design

This is a monocentric, non-interventional, retrospective study conducted at the University Hospital Essen’s ED in Germany between 2 March and 31 March 2023. Our goal was to screen all patients presenting to the emergency department. Two hundred eighty-seven patients with acute internal or neurological problems were screened during 289 ED visits (Figure 3). Verbal informed consent was obtained before the study. Inclusion criteria were adult patients with non-infectious conditions (due to their treatment in a separate area of the emergency department). Exclusion criteria included patients with language barriers and those with less severe conditions, such as young adults with a known migraine episode, as these were treated in a separate area considered pre-ED rather than being admitted to the ED itself. Patients with impaired consciousness or severe physical or psychological symptoms were also included; however, their inclusion often resulted in incomplete data due to the inability to obtain a comprehensive medical history, assess symptoms, or obtain informed consent.

Patients were interviewed, and the P-CaRES and SPICT tools were completed simultaneously using information from routine medical interviews and data from the hospital’s computerized medical records. Patient characteristics, including age, sex, diagnosis, symptoms, treatment, status of DNR (Do Not Resuscitate)/DNI (Do Not Intubate), and the presence of advanced directives, were also collected.

Moreover, 26 healthcare workers were surveyed regarding the pre-final German version of the P-CaRES tool in order to assess content and face validity.

### 2.2. Linguistic and Cultural Adaptation

Linguistic and cultural adaptation of the translation was carried out in accordance with the World Health Organization (WHO) Guidelines [37]. Two bilingual translators with clinical expertise, including one certified English translator, conducted two independent forward translations, blinded to the original tool. The translations were then evaluated and merged by an expert panel, consisting of three palliative care physicians and two palliative care nurses. A backward translation was carried out by another bilingual translator with clinical expertise, also blinded to the original tool. The draft version was reviewed by the expert panel and compared with the original version. Final adjustments were made to the pre-final German version, which was subsequently tested in the ED.

## 3. Measures

### 3.1. SPICT

The Supportive and Palliative Care Indicators Tool (SPICT) is designed to identify possible palliative care needs or general health deterioration showing moderate to high sensitivity and specificity across various studies [26,27,28,38,39]. It consists of two sections: the first section gathers subjective information about the patient’s condition and possible deterioration, while the second section inquiries about underlying illnesses. The tool was originally developed at the University of Edinburgh in 2010. We used the validated German version from 2019 [26] (Appendix A).

### 3.2. P-CaRES Tool

The P-CaRES tool is a short and precise two-part questionnaire designed to assess palliative care needs (Figure 1 and Figure 2). The first section assesses the presence of advanced underlying diseases. If any question is answered affirmatively, the second section is completed; otherwise, the screening is negative. The second section includes patient-specific questions, such as the “surprise question”, allowing the practitioner to make an individualized assessment. If at least two items in this section are positive, a specialized palliative care consultation is recommended. This tool has been tested in the USA [34,35,40,41], Thailand [42], and Switzerland [43]. Although it has been used in two German-speaking countries—Austria [44] and Germany [45]—the tool has not been validated in these contexts prior to these studies. The order in which the tools were used was neither predefined nor randomized.

### 3.3. Content and Face Validity

We assessed content and face validity using an anonymous multi-page questionnaire completed by healthcare workers in the ED. Content validity ensures the tool’s relevance, sufficiency, and clarity [46], while face validity measures whether the tool appears to cover its intended concept [47].

To assess face validity, various aspects of the tool’s design were evaluated, including comprehensibility, practicability, precision, and relevance. Respondents rated each question on a Likert scale from 1 (strongly disagree) to 5 (strongly agree), with an option to select zero points if no statement applied. The Likert scale evaluation was conducted as follows: a mean rating of ≥4 points indicated strong agreement, a rating between 2.1 and 3.9 points indicated moderate agreement, and a rating of ≤2 points indicated disagreement.

For content validity, each item was evaluated with a yes-or-no response. A cut-off for strong acceptance was set at ≥80% positive responses among all participants, while ≤50% indicated strong disagreement. Open-ended questions allowed for suggestions and feedback regarding superfluous or missing aspects. Personal data, including age, position, further training, job title, and experience were collected. Questionnaires were distributed personally in the ED and palliative care ward or sent via email.

### 3.4. Construct Validity

At the time of this study, no stablished German gold standard exists for measuring palliative care needs in the ED. Consequently, the German validated tool SPICT was used to assess construct validity by comparing the results of both tools, with SPICT serving as the imperfect gold standard. Both tools evaluate similar aspects. For comparison purposes, the same cut-off values were employed for both SPICT and P-CaRES. These values were created by imitating P-CaRES’s score system: one point was added in the disease-specific section and two points in the “palliative care needs” section. Sensitivity, specificity, and positive and negative predictive values were determined, as well as Cohen’s kappa and 95% confidence intervals (CI). The cut-off for high agreement was set at ≥80%.

### 3.5. Expert Opinion

A physician with extensive experience in specialized palliative care, blinded to the tool, independently reviewed medical data from the day of the ED visit to evaluate the necessity and appropriateness of a palliative care service consultation, comparing these results with those obtained from the P-CaRES. Sensitivity, specificity, positive predictive value, negative predictive value, as well as Cohen’s kappa and 95% CI were determined. The expert opinion served as the gold standard, with a cut-off for high agreement set at ≥80%.

### 3.6. Ethics

The study’s procedures involving human participants adhered to the ethical standards set by the institutional and local ethical review board of the University of Duisburg-Essen. The data analysis was approved under ethics vote number 23-11093-BO (30 March 2023).

## 4. Results

### 4.1. Cultural and Linguistic Adaptation and Translation of the Tool

The translation proceeded smoothly, with only minor semantic and conceptual differences requiring slight grammatical changes, leading to the pre-final German version tested in the ED. One suggestion was to separate the items “uncertainty about goals of care” and “caregiver distress” due to their differing meanings, but this was rejected to avoid altering the scoring system.

### 4.2. Recruitment of Patients in the ED

Two hundred eighty-seven patients were screened over four weeks. The mean age of the participants was 60.7 years, with a range of 19–97 years. Most common reasons for referral were “Malaise and Fatigue”, “Dyspnea”, “Abdominal Pain”, and “Chest Pain”. All patients’ characteristics are presented in Table 1. A total of 256 patients were included in the study, with 258 screenings conducted due to duplicate consultations. As screening always focuses on the patient’s current condition, which may change between consultations, a total of 289 consultations were evaluated. Two hundred fifty-six patients and 258 screenings were included in the analysis. The remaining 31 patients were excluded for various reasons, including severe cognitive impairment and intoxication, as this resulted in missing data.

Moreover, 50 patients were identified as having palliative care needs, representing 19.4% of all screened patients (Table 2).

### 4.3. Characteristics of Healthcare Workers

A total of 26 healthcare workers, actively engaged in the ED or palliative medicine at the time of the study, participated in the survey, with 65.4% completing the questionnaire in its entirety. The mean age of respondents was 40.7 years. Among the participants, 46.2% were nurses and 42.3% were medical doctors. As both emergency and palliative care healthcare workers were surveyed, only 19.2% had experience in both areas (Table 3).

### 4.4. Face Validity

The surveyed healthcare workers agreed with the tool’s design, including its comprehensibility, applicability, ease of use, precision, and relevance (Table 4). A considerable proportion of respondents strongly agreed that screening for palliative care needs represents a pivotal aspect of patient assessment in the ED, with an average rating of 4.52 points. Moreover, the desire for a palliative care consultation within 24 h was the most strongly endorsed, with an average rating of 4.56 points. In addition, participants reported non-existent use of comparable instruments in the ED, with an average rating of 1.56 points. Considering the favorable evaluations, no additional modifications were implemented in the tool.

### 4.5. Content Validity

All participants agreed that “advanced cancer” and “end-stage liver disease” are clear indicators for a specialized palliative care consultation (100%). Conversely, there was less consensus on three items: 68.0% supported “advanced renal failure” as an indicator, 66.7% found the “surprise question” relevant, and only 53.9% agreed with “frequent hospitalization” (Table 5).

### 4.6. Participants’ Suggestions for Improvement

Participants identified several potential omissions in the tool when assessing palliative care needs, such as advanced age, malnutrition, or weight loss. Certain elements, including “septic shock” and “provider discretion”, were considered possibly redundant. Feedback from healthcare workers gathered through informal interviews and comments included suggestions such as replacing “septic shock” with “shock in general” and organizing organ systems into bullet points for clarity. Opinions differed on the appropriate level of detail. Moreover, concerns were raised that goals of care often become discernible only later and not necessarily during emergencies.

### 4.7. Construct Validity

In the construct validity analysis, the majority of SPICT items showed a strong agreement with corresponding P-CaRES items (Appendix A). SPICT was employed as the benchmark for comparison in all subsequent calculations. Overall, an agreement was found in 88.4% of cases (Cohen’s kappa: 0.67, 95% CI: [0.56, 0.78]). In addition, the following statistical values were determined: specificity: 96.8%, sensitivity: 64.7%, positive predictive value: 88.0%, negative predictive value: 88.5% (Table 6).

### 4.8. Comparison with Expert’s Opinion

Overall, an agreement was found in 85.5% of cases (Cohen’s kappa: 0.54, 95% CI: [0.41, 0.68]). In addition, the following statistical values were determined: specificity: 90.8%, sensitivity: 64.0%, positive predictive value: 62.7%, negative predictive value: 91.2% (Table 7).

## 5. Discussion

In this study, a culturally and linguistically equivalent German version of the P-CaRES tool was validated for identifying patients with palliative care needs in the ED.

The cultural and linguistic adaptation and translation of the tool proceeded unproblematically. The face validity analysis demonstrated strong approval across all aspects (Table 4), particularly regarding the simplicity and acceptance of the tool, which is highly relevant given that the ED requires quick decision-making and rapid responses. These results align with current recommendations that advocate for the integration of palliative care into emergency medicine training [45,48,49].

These findings corroborate those obtained by Bowman et al. [34] during the original implementation of the tool. However, in contrast to our survey, 35% of respondents expressed doubts about the feasibility and applicability of the tool in their final working environment.

Content validity analysis showed a strong consensus regarding all aspects of identifying possible palliative care needs (Table 5). These results are similar to those of George et al. [35]. However, items such as “renal failure”, “septic shock”, and “provider discretion” showed slightly lower agreement levels in our study. Concerns were raised about the relevance of “renal failure” for dialysis patients by one healthcare worker and four healthcare workers considered this aspect to be superfluous, as reflected in the lower approval rate for this item. The lower agreement on “provider discretion” may relate to the separation between internal medicine and surgical emergency departments (see the Section Limitations), as respondents often lack experience with trauma patients, who are mentioned as examples in the tool. One employee noted that the urgent nature of emergencies might complicate subjective assessments, potentially affecting accuracy.

In the second part of the tool, three items were categorized as particularly relevant: “uncontrolled symptoms”, “functional decline”, and “uncertainty about treatment goals”, consistent with findings from the original study [34]. The item “uncertainty about treatment goals”, which was found to be highly relevant by participants in this study, highlights the importance of promoting conversations about care goals and shared decision-making, which in turn enhances well-being for both patients and caregivers [50]. The item “frequent hospitalizations” exhibited a moderate level of agreement (53.9%), aligning with the lowest agreement rate (66.7%) reported in the original study [34], despite studies indicating an increase of ED visits at the end of life [7,19,20]. In our study, this aspect was noted in 88.0% of patients who met the criteria for palliative management as defined by the tool. Additionally, Ouchi et al. [51] found a significant difference in hospital admissions between P-CaRES positive and negative patients (79% vs. 24%). These results emphasize the importance of including this item in the ED evaluation. The item “surprise question” received a moderate agreement (66.7%) in our study. Prior studies demonstrated its effectiveness [42,52,53,54], with Kirkland et al. [55] and Ouchi et al. [51] demonstrating that 80–85% of P-CaRES positive patients responded positively to the surprise question, compared to 25–35% of P-CaRES negative patients. Koyavatin et al. [42] investigated the efficacy of the “surprise question” in predicting mortality among ED patients, demonstrating its effectiveness in this regard at both one- and three-month intervals. In line with these studies, George et al. [34] reported 100% agreement among this aspect in their study.

Participant comments highlighted time constraints in ED settings, a concern also noted by Bowman et al. [35], where 35% of respondents felt their work environment could impede the tool’s usage. Interestingly, Ouchi et al. [51] reported an average screening time of 1.8 min, aligning with the rapid pace required in the ED.

The absence of a gold standard makes the analysis of construct validity challenging. The measurement of construct validity by comparing P-CaRES with the SPICT tool showed an overall agreement of 88.4% (Table 6). The high specificity and moderate sensitivity may be attributed to the fact that SPICT contains more detailed items related to the same aspects, potentially leading to quicker positive results compared to P-CaRES.

The expert agreed with the results in 85.5% of cases (Table 7), which is comparable to Bowman et al.’s [35] findings of 88.7% agreement using case vignettes. The tool’s high specificity of 90.8% and high negative predictive value of 91.2% further support its effectiveness in optimizing the use of palliative care service and avoiding unnecessary consultations. However, its sensitivity (64.0%) and positive predictive value (62.7%) were moderate. A low PPV (Table 7) can be attributed to multiple factors. Statistically, PPV depends on sensitivity, specificity, and the prevalence of the condition of interest. In this study, the tool’s sensitivity was moderate (64%) and the prevalence of unrecognized palliative care needs in the observed patient cohort was relatively low (19.5%), both of which contribute to a lower PPV. Additionally, while the tool effectively identifies patients with potential palliative care needs, it may not fully align with the nuanced clinical judgment of experts, leading to a higher rate of false positives. Another possible explanation is that the tool may detect patients in earlier stages of illness, when palliative care is not yet deemed necessary, resulting in over-identification. The implications of a low PPV should be carefully considered. An increased rate of false positives may lead to unnecessary healthcare utilization, including premature initiation of palliative care services. However, palliative care, when introduced appropriately, has been shown to improve quality of life and symptom management. It is also essential to weigh the risks of false negatives, as delayed initiation of palliative care remains a more frequent issue in clinical practice [16,17,18]. Further studies are needed to refine the tool’s threshold, balancing sensitivity, and specificity to optimize its clinical utility across different settings.

Finally, our study confirmed a high rate of unrecognized palliative care needs in ED patients, with 19.4% of screenings yielding positive results. These findings align with previous studies, which reported positive screening rates of 13.2% [44] and 33% [51]. Boyle et al. [56] also report a 400% increase in palliative care consultations after implementing evidence-based screening in the ED. Further research is needed to investigate the benefits of using P-CaRES as a screening tool in the ED and to explore various models for delivering palliative care in this setting.

### Limitations

This study has several limitations. It was conducted in a single ED that does not treat surgical patients, limiting the generalizability of our results. The overrepresentation of hematological and oncological patients from a specialized cancer center may have introduced bias (Table 2). Additionally, the exclusion of less severe cases, unaccompanied patients with cognitive impairments, and those with language barriers may have further skewed the data. The study did not evaluate interrater reliability and lacked a gold standard for tool validation. Additionally, ED physician expertise in palliative care may have influenced patient management without the need to activate palliative care services. Finally, the order in which the tools were used was neither predefined nor randomized. While we do not expect an order effect, the lack of a predefined administration sequence could be considered a minor methodological limitation.

## 6. Conclusions

A culturally and linguistically equivalent German version of the P-CaRES tool was validated, demonstrating high face- and content validity as well as high construct validity. The broader implementation of this tool in the German healthcare context could significantly benefit patients with potential palliative care needs.

## Figures and Tables

**Figure 1 jcm-14-02191-f001:**
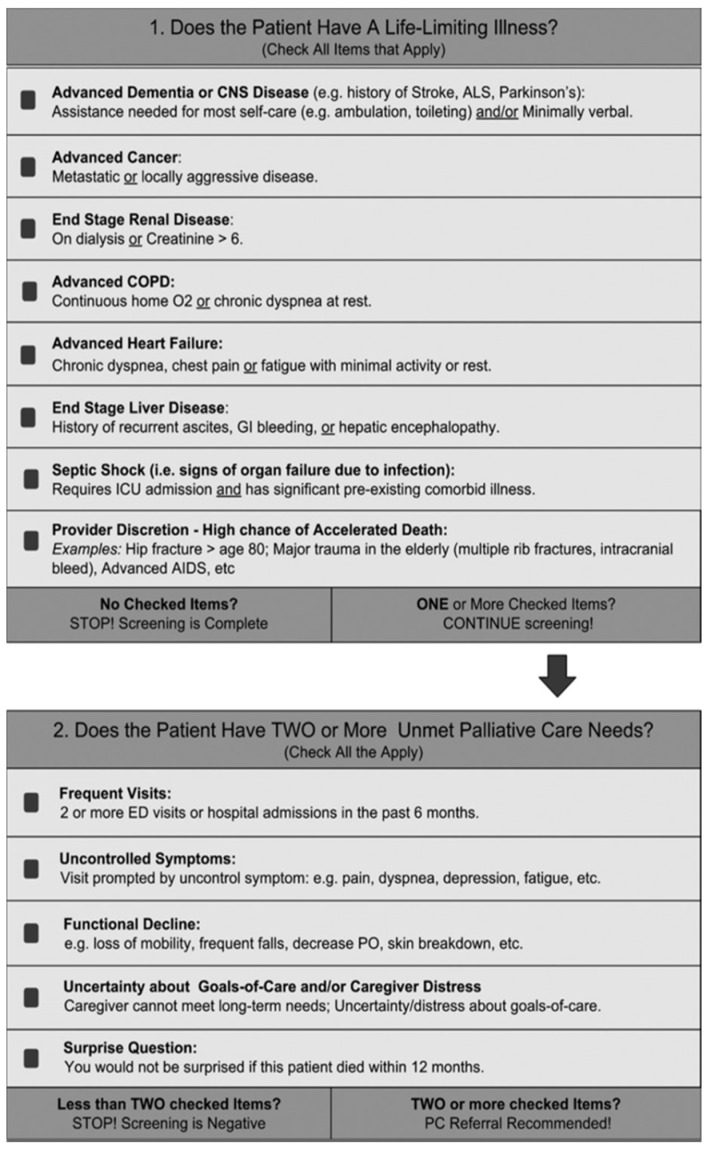
Original version of the P-CaRES tool.

**Figure 2 jcm-14-02191-f002:**
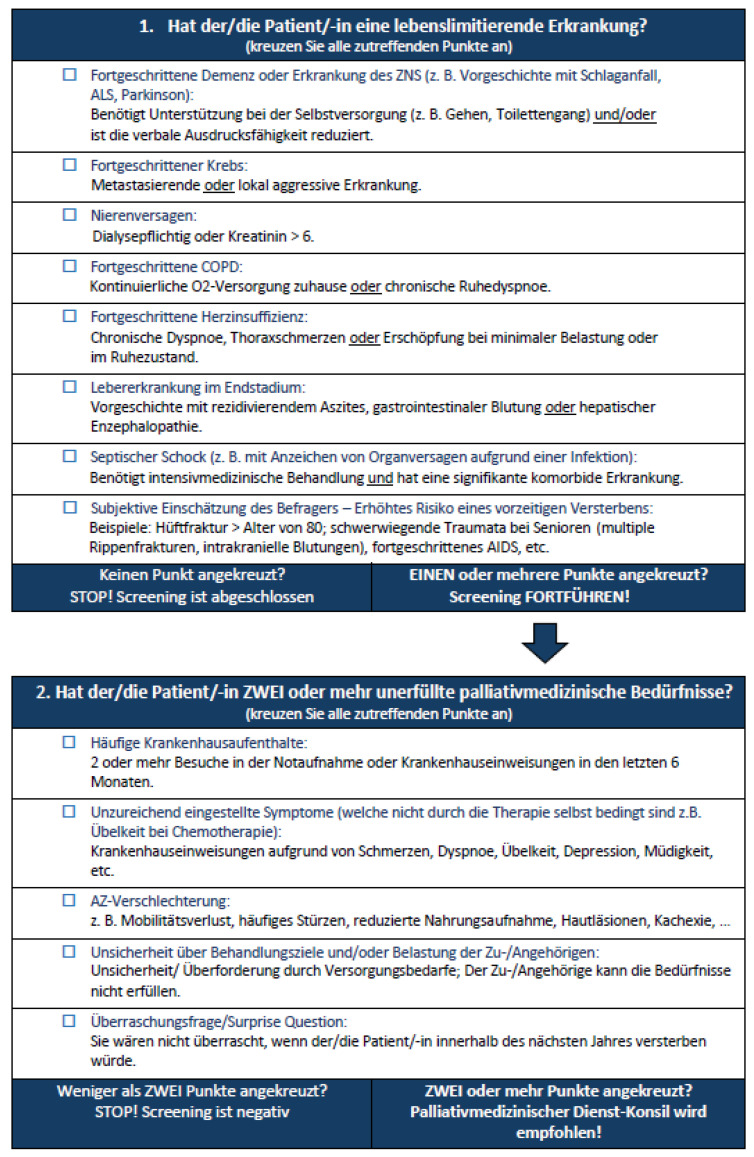
German translated and validated version of the P-CaRES tool.

**Figure 3 jcm-14-02191-f003:**
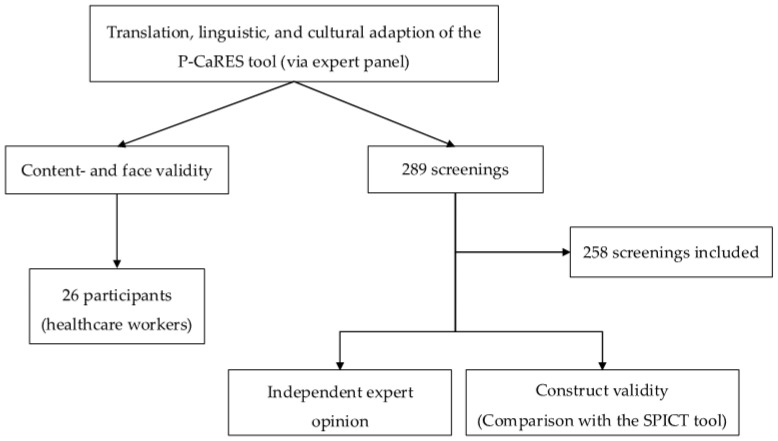
Flow chart of the methods used in the study.

**Table 1 jcm-14-02191-t001:** Patient characteristics. Characteristics of the 258 patients from the emergency department who were screened. The absolute number of patients and the percentage of the total population for each aspect are presented.

Variable		Patients/ED-Consultations (258 in Total)	
		n	%
**Age**			
	≤60	114	44.2
	>60	144	55.8
**Gender**			
	male	135	52.3
	female	123	47.7
**Existence of a living will**			
	yes	75	29.1
	no	183	70.9
**DNR/DNI status**			
	yes	17	6.6
	no	241	93.4
**Reasons for referral (most common)**			
	Malaise and Fatigue	59	22.9
	Dyspnea	31	12.0
	Abdominal Pain	29	11.2
	Chest Pain	27	10.5
	Neurological Disease	20	7.8
	Hypertension	10	3.9
	No Diagnosis Given	10	3.9
	Dizziness and Staggering	9	3.5
	Syncope/Collapse	8	3.1
	Palpitations	7	2.7
	Convulsions	7	2.7
	Symptoms of the Urinary Tract System	7	2.7

DNR: “do not resuscitate”, DNI: “do not intubate”.

**Table 2 jcm-14-02191-t002:** Overview of the completed P-CaRES items. A total of 258 emergency department patients were screened using the P-CaRES tool. A palliative care consultation is recommended if at least one positive aspect is identified in the first section and two positive aspects in the second section. If positive aspects are found in the first section, the screening process concludes and the second section is not completed.

Item Fulfilled	Number of Patients	Percentage of All Patients (Total: 258)	Percentage of Patients with Minimum One Cross in the First Part (Total: 68)	Percentage of Positive Patients (Total: 50)
First part of the P-CaRES:				
CNS Disease/Advanced Dementia	13	5.0%		26.0%
Advanced Cancer	33	12.8%		66.0%
End Stage Renal Disease	1	0.4%		2.0%
Advanced COPD	9	3.5%		18.0%
Advanced Heart Failure	8	3.1%		16.0%
End Stage Liver Disease	2	0.8%		4.0%
Septic Shock	0	0.0%		0.0%
Provider Discretion	4	1.6%		8.0%
Second part of the P-CaRES:				
Frequent Visits	44	17.1%	64.7%	88.0%
Uncontrolled Symptoms	40	15.5%	58.8%	80.0%
Functional Decline	34	13.2%	50%	68.0%
Uncertainly about Goals of Care	0	0.0%	0.0%	0.0%
Caregiver Distress	2	0.8%	2.9%	4.0%
Surprise Question	23	8.9%	33.8%	46.0%

CNS: central nervous system, COPD: chronic obstructive pulmonary disease.

**Table 3 jcm-14-02191-t003:** Characteristics of interviewed healthcare workers. Twenty-six healthcare workers from various professional groups, either employed in the local emergency department or in palliative care, were interviewed.

Role	Absolute Number and Percentage of All Participants	Age in Years (M)	Experience in PM (%)	Experience in the ED (Months)
Nurse	12 (46.2%)	43.5	58.3	59.0
Medical Doctor	11 (42.3%)	42.4	81.8	88.9
Student	2 (7.7%)	22.5	0.0	15.0
Physician Assistant	1 (3.8%)	24.0	0.0	20.0
**Total**	26	-	53.8	57.7

PM: palliative medicine, ED: emergency department, M: mean value of all participants.

**Table 4 jcm-14-02191-t004:** Face validity assessed by healthcare workers. The face validity was assessed using a five-point Likert scale, where a score of one indicated strong disagreement, five points indicated strong agreement, and 0 indicated abstention from expressing an opinion. Mean scores for each statement on the five-point Likert scale, along with the absolute number of responses, are presented in the table.

	Likert-Scale-Punctuation						
Questions	M (SD)	1	2	3	4	5	0
The structure of the tool is coherent	4.42 (0.95)	1	0	2	7	16	0
The points are clearly stated	4.46 (0.86)	0	1	3	5	17	0
The points are precisely formulated	4.04 (1.04)	0	3	4	8	11	0
The tool is clearly structured	4.35 (1.06)	0	3	2	4	17	0
I consider the tool to be relevant for identifying life-limiting conditions	4.29 (1.08)	1	1	2	6	14	1
The tool does not capture the most important factors in identifying the need for PCS	1.91 (1.31)	12	7	0	2	2	3
Screening for PCN is an important part of the assessment of a patient with advanced disease	4.52 (0.92)	1	0	1	6	17	1
Patients benefit from using the tool	4.29 (0.96)	0	1	4	4	12	5
I already use a similar tool	1.56 (1.15)	12	1	2	0	1	10
The tool can be used independently by the ED staff	4.22 (1.13)	1	1	3	5	13	3
I think the tool is feasible to use	4.04 (1.11)	1	2	3	9	11	0
The tool is useful in my daily work	4.06 (1.09)	1	0	3	6	7	9
I would use the tool in my daily work	4.33 (1.24)	1	2	1	2	15	5
I am in favor of introducing the tool in our ED	4.22 (1.09)	1	1	2	7	12	2
If a validated screening tool indicates PCN for my patient, I would like a PCS consultation within 24 h	4.56 (1.00)	1	1	0	4	19	1

Scoring system: 1 point: disagreement, 5 points: agreement, 0 points: no statement given. M: mean value of all participants, SD: Standard deviation, PCS: palliative care service, PCN: palliative care needs, ED: emergency department.

**Table 5 jcm-14-02191-t005:** Content validity assessed by healthcare workers. Participants who independently evaluated the relevance of each item in the P-CaRES tool assessed content validity. Each item was rated with a yes-or-no response. A cut-off of 80% indicated strong acceptance, while a cut-off of 50% indicated strong disagreement.

Items of P-CaRES	Agreement M (%)
CNS Disease/Advanced Dementia	96.2
Advanced Cancer	100
End Stage Renal Disease	68.0
Advanced COPD	95.8
Advanced Heart Failure	96.0
End Stage Liver Disease	100
Septic Shock	84.0
Provider Discretion	77.3
Frequent Visits	53.9
Uncontrolled Symptoms	80.8
Functional Decline	92.0
Uncertainly about Goals of Care/Caregiver Distress	80.8
Surprise Question	66.7

M: mean value of all participants, CNS: central nervous system, COPD: chronic obstructive pulmonary disease.

**Table 6 jcm-14-02191-t006:** Construct validity of the P-CaRES tool. The construct validity was determined by comparing the results of the P-CaRES with those of the SPICT tool, including both absolute numbers and percentage of all screened patients.

		SPICT Result			
		Negative	Positive	Total	
**P-CaRES result**	Negative	184 (71.3%)	24 (9.3%)	208 (80.6%)	NPV: 88.5%
	Positive	6 (2.3%)	44 (17.1%)	50 (19.4%)	PPV: 88.0%
	total	190 (73.6%)	68 (26.4%)	258 (100%)	
		SP: 96.8%	SN: 64.7%		

Screening negative: no unmet palliative care needs. Screening positive: possible unmet palliative care needs. SP: specificity, SN: sensitivity, NPV: negative predictive value, PPV: positive predictive value.

**Table 7 jcm-14-02191-t007:** Comparison with the expert’s opinion. The results of the P-CaRES tool were compared with the opinions of an independent expert. The absolute numbers and percentage of all screened patients are presented in the table.

		P-CaRES Results			
		Negative	Positive	Total	
**Expert´s opinion**	negative	187 (73.0%)	19 (7.4%)	206 (80.5%)	NPV: 91.2%
	positive	18 (7.0%)	32 (12.5%)	50 (19.5%)	PPV: 62.7%
	total	205 (80.1%)	51 (19.9%)	256 (100%)	
		SP: 90.8%	SN: 64.0%		

Screening negative: no unmet palliative care needs. Screening positive: possible unmet palliative care needs. SP: specificity, SN: sensitivity, NPV: negative predictive value, PPV: positive predictive value.

## Data Availability

The data presented in this study are available on request from the corresponding author. The data are not publicly available due to privacy and ethical considerations.

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
