# Peer review of "Validation of the Palliative Care and Rapid Emergency Screening (P-CaRES) Tool in Germany"

_jcm, 2025, doi:10.3390/jcm14072191_

Round 1

Reviewer 1 Report

Comments and Suggestions for Authors

I appreciate the opportunity to review “Validation of the Palliative Care and Rapid Emergency Screening (P-CaRES) Tool in Germany” by Schmitz et al. and thank the Editor for this invitation.

In this study, the authors aimed to translate, adapt, and validate the P-CaRES tool for a different healthcare context.

I found their investigation to be both relevant and insightful. However, I would like to raise the following concerns and suggestions for improvement, presented in no particular order of importance:

SPECIFIC COMMENTS

  • p. 3, L86-88: I recommend that the authors acknowledge that timely referral to palliative care (PC) consultations may help alleviate patients' spiritual distress as well, in addition to the aspects they have already mentioned (i.e., QoL, mood, symptom management, prolonging life expectancy). In this regards, strong evidence supports that PC both mitigates spiritual distress and inherently integrates spiritual care as a core component of its holistic approach. Expanding the introduction to include this perspective would strengthen the manuscript. Please cite and incorporate the following reference: PMID: 39093919.
  • p. 3, L97-99: The discussion on Emergency Department (ED) admissions should not be limited to cancer patients alone, as individuals with serious chronic illnesses (e.g., COPD) face similar challenges. Research indicates that PC can also reduce ED visits for non-cancer populations. A broader perspective encompassing both cancer and non-cancer patients would provide a more balanced overview. Please address this by citing PMID: 39794226 and incorporating the findings into your discussion.
  • p. 4, L133 onwards: Given that this study is a monocentric, non-interventional, retrospective investigation, it is this expert Reviewer’s opinion that it should adhere to established international guidelines for reporting retrospective studies (e.g., STROBE, CONSORT). However, this does not appear to have been explicitly followed. I strongly recommend that the authors ensure compliance with these guidelines and provide a completed checklist as supplementary material.
  • p. 13, L458-460: The statement ‘[…] uncertainty about treatment goals’, consistent with findings from the original study [33]’ could benefit from further elaboration. Specifically, I would encourage the authors to discuss the potential impact of enhancing patients’ prognostic awareness (PA) through ongoing conversations about goals of care and shared decision-making (SDM) involving family members and caregivers. Evidence suggests that improving PA fosters well-being and hope for both patients and caregivers. Please expand on the role of family involvement in prognostication discussions and cite PMID: 39807018 to support this point.

I appreciate the authors’ work on this important topic.

Author Response

For research article manuscript jcm-3512769

Response to Reviewers:

We thank the reviewers for taking the time to review our manuscript and to provide their insightful comments. We have addressed all the reviewers' comments, and we believe that our paper has become significantly stronger as a result. Please find our detailed responses below, with the corresponding revisions and corrections highlighted in yellow in the re-submitted files. The individual responses to the reviewers' comments and the changes made to the manuscript are listed separately for reviewers R1 and R2.

Response to R1:

  1. 3, L86-88: I recommend that the authors acknowledge that timely referral to palliative care (PC) consultations may help alleviate patients' spiritual distress as well, in addition to the aspects they have already mentioned (i.e., QoL, mood, symptom management, prolonging life expectancy). In this regards, strong evidence supports that PC both mitigates spiritual distress and inherently integrates spiritual care as a core component of its holistic approach. Expanding the introduction to include this perspective would strengthen the manuscript. Please cite and incorporate the following reference: PMID: 39093919.

Thank you for this suggestion. We have included the recommended reference and accordingly changed the text.

  1. 3, L97-99: The discussion on Emergency Department (ED) admissions should not be limited to cancer patients alone, as individuals with serious chronic illnesses (e.g., COPD) face similar challenges. Research indicates that PC can also reduce ED visits for non-cancer populations. A broader perspective encompassing both cancer and non-cancer patients would provide a more balanced overview. Please address this by citing PMID: 39794226 and incorporating the findings into your discussion.

Thank you for this valuable comment. We have revised and changed the text to clarify that we have also included studies involving non-cancer populations.

  1. 4, L133 onwards: Given that this study is a monocentric, non-interventional, retrospective investigation, it is this expert Reviewer’s opinion that it should adhere to established international guidelines for reporting retrospective studies (e.g., STROBE, CONSORT). However, this does not appear to have been explicitly followed. I strongly recommend that the authors ensure compliance with these guidelines and provide a completed checklist as supplementary material.

Thank you for your valuable feedback. We appreciate your suggestion regarding adherence to established reporting guidelines. However, we would like to clarify that our study is a validation study of a screening tool in palliative patients, rather than a purely observational or interventional study. As such, while STROBE and CONSORT are important frameworks for observational and randomized studies, they may not be entirely applicable to our study design.

Instead, validation studies of screening tools typically adhere to STARD (Standards for Reporting Diagnostic Accuracy Studies), which provides specific recommendations for reporting diagnostic and screening validation research. We have ensured that our study follows these standards and would be happy to provide a completed STARD checklist as supplementary material if required.

Please let us know if you have any further recommendations or if you would prefer additional clarifications regarding our methodology.

  1. 13, L458-460: The statement ‘[…] uncertainty about treatment goals’, consistent with findings from the original study [33]’ could benefit from further elaboration. Specifically, I would encourage the authors to discuss the potential impact of enhancing patients’ prognostic awareness (PA) through ongoing conversations about goals of care and shared decision-making (SDM) involving family members and caregivers. Evidence suggests that improving PA fosters well-being and hope for both patients and caregivers. Please expand on the role of family involvement in prognostication discussions and cite PMID: 39807018 to support this point.

Thank you for this suggestion. We have included the recommended reference and accordingly changed the text.

Reviewer 2 Report

Comments and Suggestions for Authors

The authors conducted an observational study to translate, adapt, and validate the P-CaRES tool for the Deutsch healthcare context. The P-CaRES tool is designed specifically to help the Emergency physician identify patients who might benefit from palliative care referral in the ED. Suggestions:

1) In the exclusion criteria, please clarify the definition of "those with less severe conditions" and consider providing examples.

2) Lines 185-192: For SPICT please provide validation details (specificity, sensitivity, validity, reliability).

3) The manuscript mentions supplemental figures (lines 191-192), but supplemental file was not uploaded/provided for review. Please include figures for SPICT similar to fig. 1 and 2.

4) Was the sequence of administration of SPICT and P-CaRES the same in all participants? Which one was administered first? Or was it randomly shuffled and not administered in any fixed order? Can the response to one instrument introduce bias in responses to the other instrument? Please clarify in methods section and discuss potential bias in limitations section.

5) The PPV when compared to expert opinion is low (Table 7). Please discuss potential reasons and implications of a low PPV (excess unnecessary healthcare utilization).

6) For truly evaluating the tool, it is essential to compare performance using mortality/survival as an outcome, to know if palliative care is truly essential  or whether it may unnecessarily escalate care leading to additional avoidable resource utilization .

7) Implications of false positive result on patient outcomes should also be discussed- for example, whether initiating palliative care may alter course of treatment and impact on patient's health outcomes and quality of life. 

Author Response

For research article manuscript jcm-3512769

Response to Reviewers:

We thank the reviewers for taking the time to review our manuscript and to provide their insightful comments. We have addressed all the reviewers' comments, and we believe that our paper has become significantly stronger as a result. Please find our detailed responses below, with the corresponding revisions and corrections highlighted in yellow in the re-submitted files. The individual responses to the reviewers' comments and the changes made to the manuscript are listed separately for reviewers R1 and R2.

Response to R2:

  • In the exclusion criteria, please clarify the definition of "those with less severe conditions" and consider providing examples.

Thank you for pointing this out. We have modified the text (Page 4)

  • Lines 185-192: For SPICT please provide validation details (specificity, sensitivity, validity, reliability).

Thank you for this suggestion. We have provided details of the SPICT Tool in the main text (page 5)

  • The manuscript mentions supplemental figures (lines 191-192), but supplemental file was not uploaded/provided for review. Please include figures for SPICT similar to fig. 1 and 2.

We apologize for this inconvenient. We had uploaded a separate file named “Supplemental Material” which should be available to all reviewers. However, we are happy to provide this material again in case any issues have occurred.

  • Was the sequence of administration of SPICT and P-CaRES the same in all participants? Which one was administered first? Or was it randomly shuffled and not administered in any fixed order? Can the response to one instrument introduce bias in responses to the other instrument? Please clarify in methods section and discuss potential bias in limitations section.

Thank you for your thoughtful comment. In our study, we did not follow a specific sequence for administering SPICT and P-CaRES; the order of use was not predefined or randomized. Importantly, the tools were completed by healthcare providers, and results were not disclosed to patients. Given that these instruments are designed for provider-based assessment rather than patient self-report, we do not believe that the sequence of administration could have introduced bias in our results.

To address your concern, we have clarified this point in the Methods section (page 6) and explicitly stated in the Limitations section (page 14) that, while we do not anticipate an order effect, the absence of a predefined administration sequence could be considered a minor methodological limitation.

  • The PPV when compared to expert opinion is low (Table 7). Please discuss potential reasons and implications of a low PPV (excess unnecessary healthcare utilization).

Thank you for this valuable suggestion. On page 13, we have expanded the discussion on the low PPV to enhance its reach and depth.

  • For truly evaluating the tool, it is essential to compare performance using mortality/survival as an outcome, to know if palliative care is truly essential  or whether it may unnecessarily escalate care leading to additional avoidable resource utilization.

We agree that evaluating the tool in relation to mortality and survival outcomes is important to fully assess its impact on palliative care. This is why our research group is conducting a follow-up study that will specifically examine mortality as an outcome, which will help clarify this crucial aspect of the tool’s effectiveness.

In the current study, our focus is on validating a tool that has already been validated in the USA. As such, we believe that incorporating mortality outcomes is beyond the scope of the present research. However, we acknowledge the importance of this outcome and we appreciate your valuable feedback.

  • Implications of false positive result on patient outcomes should also be discussed- for example, whether initiating palliative care may alter course of treatment and impact on patient's health outcomes and quality of life. 

Thank you for your thoughtful comment regarding the implications of false positive results on patient outcomes. A false positive result from the tool could indeed lead to the premature initiation of palliative care. However, it is important to note that when initiated appropriately, palliative care has been shown to improve quality of life and symptom management, particularly in the context of serious illness. Moreover, it is equally important to consider the risks of not initiating palliative care in a timely manner, which, in practice, occurs more frequently. We have modified the text on page 13 and 14.

Round 2

Reviewer 1 Report

Comments and Suggestions for Authors

I thank the Editor and the Authors for allowing me an opportunity to review the revised version of the manuscript ‘Validation of the Palliative Care and Rapid Emergency Screening (P-CaRES) Tool in Germany’ by Schmitz et al.

First off, I would like to thank the authors for their outstanding revision effort, which has been - in this Reviewer's opinion - really beneficial.

All of this Reviewer's comments have now been adequately addressed in this rebuttal, resulting in a much stronger and more in-depth analysis of this interesting topic, which will definitely add value to this Journal.